

# Spontaneous breaking of finite group symmetries at all temperatures

Pedro Liendo[1⋆], Junchen Rong[2†] and Haoyu Zhang[3‡]

**1** DESY Hamburg, Theory Group, Notkestraße 85, D-22607 Hamburg, Germany
**2** Institut des Hautes Études Scientifiques, 91440 Bures-sur-Yvette, France
**3** George P. & Cynthia Woods Mitchell Institute for Fundamental Physics and Astronomy,
Texas A&M University, College Station, TX 77843, USA

⋆ pedro.liendo@desy.de , † junchenrong@ihes.fr ,
‡ zhanghaoyu@tamu.edu

## Abstract

We study conformal field theories with finite group symmetries with spontaneous symmetry breaking (SSB) phases which persist at all temperatures. We work with two $\lambda\phi^4$ theories coupled through their mass terms. The two $\lambda\phi^4$ theories are chosen to preserve either the cubic symmetry group, or the tetrahedral symmetry group. A one-loop calculation in the $4-\epsilon$ expansion shows that there exist infinitely many fixed points that can host an all temperature SSB phase. An analysis of the renormalization group (RG) stability matrix of these fixed points, reveals that their spectrum contains at least three relevant operators. In other words, these fixed points are tetracritical points.



# 1 Introduction

The study of spontaneous symmetry breaking phases that persist at all temperatures has recently been revisited [1,2], following the initial work of Weinberg almost fifty years ago [3]. These SSB phases are described by conformal field theories (CFTs) at finite temperature, and since the temperature is the only one scale that these systems have, the physics at high temperature and low temperature are related by scaling symmetry, which means that they are in the same phase. Another interesting feature is that near the CFT fixed points, these systems have phase diagrams with the ordered phase at high temperature, while the disordered phase is at low temperature, which appear to violate the second law of thermal dynamics. Such exotic inverted phase transitions were observed in Rochelle or Seignette salt [3,4].

In this note we consider scalar field theory with a $CFT_1 \times CFT_2$ structure. More precisely, we consider two scalar field theories ($CFT_1$ and $CFT_2$) coupled through their mass term.

$$\mathcal{L} = \mathcal{L}_{CFT_1}(\vec{\phi}_1) + \mathcal{L}_{CFT_2}(\vec{\phi}_2) + g(\vec{\phi}_1 \cdot \vec{\phi}_1)(\vec{\phi}_2 \cdot \vec{\phi}_2). \tag{1}$$

For example, take $CFT_1$ to be the O(m) invariant Wilson-Fisher CFT, and $CFT_2$ to be the O(N-m) invariant Wilson-Fisher CFT, we have the Lagrangian,

$$\begin{aligned}
\mathcal{L}_{CFT_1}(\vec{\phi}_1) &= \frac{1}{2}(\partial_\mu \vec{\phi}_1) \cdot (\partial_\mu \vec{\phi}_1) + m_1 \vec{\phi}_1 \cdot \vec{\phi}_1 + \lambda_1 (\vec{\phi}_1 \cdot \vec{\phi}_1)^2, \\
\mathcal{L}_{CFT_2}(\vec{\phi}_1) &= \frac{1}{2}(\partial_\mu \vec{\phi}_2) \cdot (\partial_\mu \vec{\phi}_2) + m_2 \vec{\phi}_2 \cdot \vec{\phi}_2 + \lambda_2 (\vec{\phi}_2 \cdot \vec{\phi}_2)^2.
\end{aligned} \tag{2}$$

Here $\vec{\phi}_1$ and $\vec{\phi}_2$ to are m-dimensional and (N-m)-dimensional vectors respectively. This is precisely the O(m)×O(N-m) theory considered in [2]. Using both the $4-\epsilon$ expansion and large-N methods, they show that these conformal field theories have a spontaneous symmetry breaking (SSB) phase that persists until infinite temperature. Setting $\epsilon = 1$, the corresponding conformal field theories are 2+1 dimensional quantum critical points. As noted already in [2], the all temperature SSB phase should be view as an SSB phase in two spatial dimensions while the temporal dimension is compact with radius $\beta = \frac{1}{T}$. The Coleman-Hohenberg-Mermin-Wagner theorem theorem [5–7] forbids spontaneous breaking of continuous symmetry in two dimension. In other words, the all temperature SSB phase exist only in fractional dimensions.

In order to bypass the Coleman-Hohenberg-Mermin-Wagner theorem, one way to proceed is to work with long range models which is the approach taken in [8,9]. Another possible direction is to work with four dimensional gauge theories [10,11].

In this note, we work with short-range models with *finite* symmetry groups. It is known that there are families of Wilson-Fisher fixed points in $4-\epsilon$ dimensions with finite groups as their global symmetry groups. For a summary of known fixed point see for example [12]. The groups we will consider here are the cubic groups $S_N \ltimes (Z_2)^N$ [13–15] and the tetrahedral groups $S_{N+1} \times Z_2$ [16]. We will call these CFTs "Cubic(N) CFTs" and "Tetrahedral(N) CFTs". We take the $CFT_1$ and $CFT_2$ in (1) to be these finite group CFTs. As explained in [2,3], searching for all temperature SSB phases corresponds to search for conformal fixed points, whose thermal mass squared matrix

$$M_{ij}^2 = \frac{\beta^{-2}}{24} \lambda_{ijkk} \Big|_{\lambda = \lambda^*} \tag{3}$$

has negative eigenvalues. This formula comes from the one loop renormalization of the zero mode mass due to the Kaluza-Klein modes along the temporal circle. The coupling constant is

defined through the relation

$$V(\phi) = \frac{1}{4!}\lambda_{ijkl}\phi^i\phi^j\phi^k\phi^l, \tag{4}$$

where we have used $\phi^i$ to collectively denote both $\vec{\phi}_1$ and $\vec{\phi}_2$.

In real life experiments, relevant operators of a conformal field theory that are invariant under all flavor symmetry groups correspond to experimental parameters that need to be fine-tuned [17]. It is therefore desirable to search for conformal field theories with as little relevant operators as possible. Notice the group O(m)×O(N-m) has two quadratic invariants, which are the two mass terms $(\vec{\phi}_1 \cdot \vec{\phi}_1)$ and $(\vec{\phi}_2 \cdot \vec{\phi}_2)$. These mass operators are relevant. One may wonder if it is possible to consider scalar field theories whose flavor symmetry allow only one mass term. It is shown in [2] that these models can not host all temperature SSB phases (at least perturbatively). Additional relevant operator may come from the quadratic $\phi^4$ terms. Following the convention of [2], the one loop beta function for the quartic couplings is given by

$$\beta_{ijkl}(\lambda_{ijkl}) = -\epsilon\lambda_{ijkl} + \frac{1}{16\pi^2}(\lambda_{ijmn}\lambda_{mnkl} + 2 \text{ permutations}). \tag{5}$$

The conformal fixed points are given by

$$\beta_{ijkl}(\lambda_{ijkl}) = 0. \tag{6}$$

Due to symmetry, the quartic couplings $\lambda_{ijkl}$ are not independent. Denote the independent couplings as $\lambda_i$, then the stability matrix of the RG flow can be written as

$$H_{ij} = \left.\frac{\partial\beta_i}{\partial\lambda_j}\right|_{\lambda=\lambda^*}. \tag{7}$$

The negative eigenvalues of $H_{ij}$ correspond to relevant operators. In looking for CFTs with a lower number of relevant operators, we will calculate the renormalization group stability matrix $H_{ij}$ of these fixed points. The result, however, shows that for all the candidate CFTs that can host all temperature SSB phases, their $H_{ij}$ matrix have at least one negative eigenvalue. A similar result was already notice in [2] for the O(m)×O(N-m) model. This means that the all temperature SSB phases appear near tetracritical points of the their phase diagrams, which can make the search for these phases in real life or computer experiments challenging. One example of such tetracritical points is the tetracritical Ising model, which is also the two dimensional minimal model $M_{6,5}$ [18–20] with central charge $c = 4/5$. Another interesting tetracritical point is the O(5) invariant Wilson Fisher fixed point as discussed in [21, 22]. In these studies, the lattice model preserves the SO(2)×SO(3) subgroup of SO(5). In view of this subgroup, the O(5) Wilson Fisher fixed point contains three relevant operators that are invariant under the lattice symmetry, they are $O_1 = \phi_1^2 + \phi_2^2$, $O_2 = \phi_3^2 + \phi_4^2 + \phi_5^2$ and $O_3 = (\phi_1^2 + \phi_2^2)(\phi_3^2 + \phi_4^2 + \phi_5^2)$. This model interestingly describes the intervening between super-fluid phase and the anti-ferromagnetic phase. For a review of this model, see also Section 6.2 of [17].

## 2 Coupled CFTs with finite group symmetries

For convenience, we scale out $\frac{1}{16\pi^2}$ and $\epsilon$ in the beta function (9) by redefining

$$\tilde{\lambda} = \frac{\lambda}{16\pi^2\epsilon}. \tag{8}$$

The condition that $\beta_{ijkl} = 0$ becomes

$$\tilde{\lambda}_{ijkl} = \tilde{\lambda}_{ijmn}\tilde{\lambda}_{mnkl} + 2\,\text{permutations}\,. \tag{9}$$

In the following we will drop the tilde for convenience.

## 2.1  Cubic(M) × Cubic(N)

As already mentioned, the Cubic(N) group is isomorphic to $S_N \ltimes (Z_2)^N$. The N copies of the $Z_2$ subgroup flip the signs of the $\phi_i$ fields, while the $S_N$ group permutes them. There are two invariant tensors $\delta_{\mu\nu}$ and

$$\delta_{\mu\nu\rho\sigma} = \begin{cases} 1\,, & \text{if } \mu = \nu = \rho = \sigma\,, \\ 0\,, & \text{otherwise}\,. \end{cases} \tag{10}$$

We take both CFT$_1$ and CFT$_2$ to be the Cubic groups. The coupling constants are

$$\lambda^{(1)}_{\mu\nu\rho\sigma} = \lambda_1(\delta_{\mu\nu}\delta_{\rho\sigma} + \delta_{\mu\rho}\delta_{\nu\sigma} + \delta_{\mu\sigma}\delta_{\nu\rho}) + \lambda_2\delta_{\mu\nu\rho\sigma}\,, \tag{11}$$

$$\lambda^{(2)}_{ijkl} = \lambda_3(\delta_{ij}\delta_{kl} + \delta_{ik}\delta_{jl} + \delta_{ijl}\delta_{jk}) + \lambda_4\delta_{ijkl}\,, \tag{12}$$

$$\lambda^{(c)}_{\mu\nu ij} = \lambda_5\delta_{\mu\nu}\delta_{ij}\,, \tag{13}$$

where $\lambda_{\mu i\nu j}, \lambda_{\mu ij\nu}\cdots$ are fixed by permutations.

According to the group structure, the beta function $\beta_{IJKL}$ can be written as

$$\begin{aligned}\beta_{IJKL} = {}& \beta_1\left(\delta_{\mu\nu}\delta_{\rho\sigma} + \delta_{\mu\rho}\delta_{\nu\sigma} + \delta_{\mu\sigma}\delta_{\nu\rho}\right) + \beta_2\delta_{\mu\nu\rho\sigma} \\ & + \beta_3\left(\delta_{ij}\delta_{kl} + \delta_{ik}\delta_{jl} + \delta_{il}\delta_{jk}\right) + \beta_4\delta_{ijkl} + \beta_5\left(\delta_{\mu\nu}\delta_{ij} + \cdots\right)\,.\end{aligned} \tag{14}$$

From (9), we can read off the $\beta_i$'s,

$$\beta_1 = -\lambda_1 + 8\lambda_1^2 + 2\lambda_2\lambda_1 + \lambda_1^2 M + \lambda_5^2 N\,, \tag{15}$$

$$\beta_2 = -\lambda_2 + 3\lambda_2^2 + 12\lambda_1\lambda_2\,, \tag{16}$$

$$\beta_3 = -\lambda_3 + 8\lambda_3^2 + 2\lambda_4\lambda_3 + \lambda_3^2 N + \lambda_5^2 M\,, \tag{17}$$

$$\beta_4 = -\lambda_4 + 3\lambda_4^2 + 12\lambda_3\lambda_4\,, \tag{18}$$

$$\beta_5 = \lambda_5\left(\lambda_2 + 2\lambda_3 + \lambda_4 + 4\lambda_5 + \lambda_1(M+2) + \lambda_3 N - 1\right)\,. \tag{19}$$

Solving for fixed points and plugging the fixed points couplings into the thermal mass matrix, we can examine the eigenvalues of $M_{ij}$ case by case . For a specific choice of M, there is an infinite family of choices of N, which leads to CFTs with all temperature SSBs. The results are summarized in Fig. 1. The linear growth of the plot at large N can be seen from a large N analysis (with M/N fixed). The details are summarized in Appendix A. It's always the symmetry group with smaller order that gets spontaneously broken. The RG stability matrix $H_{ij}$ of these fixed points have at least one negative eigenvalue. Together with the two mass terms, this means that these CFTs have at least three relevant operators (which are invariant under the flavor symmetry groups).

It is not particularly illuminating to present all the details of the solutions of (15). To demonstrate our result, we list in Table 1 the coupling constants, mass matrix eigenvalues and the RG stability matrix eigenvalues in the Cubic(60)×Cubic(4) case. We have omitted the fixed points when at least one of the coupling constants vanishes. They correspond either to CFTs when the symmetry gets enhanced to continuous groups, or CFTs when CFT$_1$ and CFT$_2$ are decoupled.

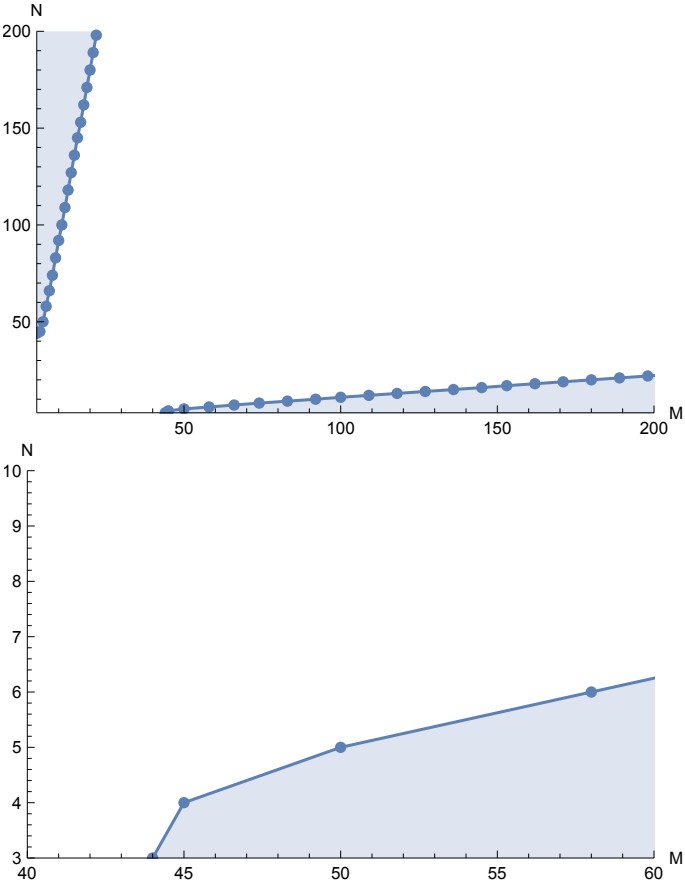

Figure 1: Cubic(M)×Cubic(N). The right panel is a zoom in of the left panel plot.

## 2.2 Tetrahedral(M)×Tetrahedral(N)

The Tetrahedral group is simply the product of the permutation group $S_{N+1}$ and the cylic group $Z_2$. The scalar fields transform in the standard (N dimensional) representation of the group $S_{N+1}$. It is well known that in this representation, the group $S_{N+1}$ has two invariant tensors $\delta_{\mu\nu}$ and $d_{\mu\nu\rho}$ which is fully symmetric and traceless. The explicit construction of $d_{\mu\nu\rho}$ can be found in [16]. The independent coupling constants can be defined as

$$\lambda_{\mu\nu\rho\sigma}^{(1)} = \lambda_1 \left( \delta_{\mu\nu}\delta_{\rho\sigma} + \delta_{\mu\rho}\delta_{\nu\sigma} + \delta_{\mu\sigma}\delta_{\nu\rho} \right) + \lambda_2 \left( d_{\mu\nu\kappa}d_{\kappa\rho\sigma} + d_{\mu\rho\kappa}d_{\kappa\nu\sigma} + d_{\mu\kappa\sigma}d_{\sigma\rho\nu} \right), \quad (20)$$

$$\lambda_{ijkl}^{(2)} = \lambda_3 \left( \delta_{ij}\delta_{kl} + \delta_{ik}\delta_{jl} + \delta_{il}\delta_{jk} \right) + \lambda_4 \left( d_{ijm}d_{mkl} + d_{ikm}d_{mjl} + d_{ilm}d_{mkj} \right), \quad (21)$$

$$\lambda_{\mu\nu ij}^{(c)} = \lambda_5 \delta_{\mu\nu}\delta_{ij}, \quad (22)$$

where $\lambda_{\mu i\nu j}, \lambda_{\mu ij\nu} \cdots$ are fixed by permutation. From

$$\beta_{IJKL} = \beta_1 \left( \delta_{\mu\nu}\delta_{\rho\sigma} + \delta_{\mu\rho}\delta_{\nu\sigma} + \delta_{\mu\sigma}\delta_{\nu\rho} \right) + \beta_2 \left( d_{\mu\nu\kappa}d_{\kappa\rho\sigma} + d_{\mu\rho\kappa}d_{\kappa\nu\sigma} + d_{\mu\sigma\kappa}d_{\kappa\rho\nu} \right) \quad (23)$$

$$+ \beta_3 \left( \delta_{ij}\delta_{kl} + \delta_{ik}\delta_{jl} + \delta_{il}\delta_{jk} \right) + \beta_4 \left( d_{ijm}d_{mkl} + d_{ikm}d_{mjl} + d_{ilm}d_{mkj} \right) \quad (24)$$

$$+ \beta_5 \left( \delta_{\mu\nu}\delta_{ij} + \cdots \right), \quad (25)$$

Table 1: Coupling constants, mass matrix eigenvalues and the RG stability matrix eigenvalues of the Cubic(60)×Cubic(4) fixed points. Here $\lambda_i$ are the couplings at the critical point, $m_i$ and $H_i$ are the eigen-values of $M_{ij}$ and $H_{ij}$ respectively. The $F_1$ fixed point has negative $m_2$, therefore can host all temperature SSB phase.

| fixed points | $\lambda_1$ | $\lambda_2$ | $\lambda_3$ | $\lambda_4$ | $\lambda_5$ | $m_1/(\frac{2}{3}\pi^2\epsilon\beta_{\text{th}}^2)$ | $m_2/(\frac{2}{3}\pi^2\epsilon\beta_{\text{th}}^2)$ |
|---|---|---|---|---|---|---|---|
| $F_1$ | 0.00420718 | 0.316505 | 0.0631077 | 0.0809026 | -0.00922458 | 0.540451 | -0.0939264 |
| $F_2$ | 0.00330785 | 0.320102 | 0.0496178 | 0.134862 | 0.0105606 | 0.567431 | 1.0662 |
| $F_3$ | 0.00520833 | 0.3125 | 0.00520833 | 0.3125 | 0.00520833 | 0.65625 | 0.65625 |

| fixed points | $H_1$ | $H_2$ | $H_3$ | $H_4$ | $H_5$ |
|---|---|---|---|---|---|
| $F_1$ | 1. | 0.929986 | 0.259747 | -0.205562 | 0.0527271 |
| $F_2$ | 1. | 0.914259 | 0.24244 | -0.23492 | 0.0359878 |
| $F_3$ | 1. | 0.979167 | -0.3125 | 0.3125 | 0 |

and (9), we can read out the beta functions

$$\beta_1 = -\lambda_1 + \frac{4\lambda_2^2(M-1)(M+1)^4}{M^6} + \lambda_1\left(\frac{4\lambda_2(M-1)(M+1)^2}{M^3}\right) + \lambda_1^2(M+8) + \lambda_5^2 N, \quad (26)$$

$$\beta_2 = -\lambda_2 + 12\lambda_1\lambda_2 + \frac{3\lambda_2^2(M+1)^2(3M-7)}{M^3}, \quad (27)$$

$$\beta_3 = -\lambda_3 + \lambda_5^2 M + \frac{4\lambda_4^2(N-1)(N+1)^4}{N^6} + \lambda_3\left(\frac{4\lambda_4(N-1)(N+1)^2}{N^3}\right) + \lambda_3^2(N+8), \quad (28)$$

$$\beta_4 = -\lambda_4 + 12\lambda_3\lambda_4 + \frac{3\lambda_4^2(N+1)^2(3N-7)}{N^3}, \quad (29)$$

$$\beta_5 = -\lambda_5 + 4\lambda_5^2 + \frac{2\lambda_2\lambda_5(M-1)(M+1)^2}{M^3} + \frac{2\lambda_4\lambda_5(N-1)(N+1)^2}{N^3} \quad (30)$$

$$+ \lambda_3\lambda_5(N+2) + \lambda_1\lambda_5(M+2). \quad (31)$$

From our numerical search, we find that all temperature SSB phases appear only when $M = 3$ and $N \geq 51$ (or when $N = 3$ and $M \geq 51$). Notice the Tetrahedral(3) group is isomorphic to the Cubic(3) group. We will consider the Cubic(M)×Tetrahedral(N) CFTs in the next section. Overall, all the CFTs we found to exhibit persistent symmetry breaking have at least one symmetric group which is the Cubic group. The RG stability matrix $H_{ij}$ of these fixed points have at least one negative eigen-values.

## 2.3 Cubic(M)×Tetrahedral(N)

In the cases of Cubic(M)×Tetrahedral(N), we define the coupling constant as

$$\lambda_{\mu\nu\rho\sigma}^{(1)} = \lambda_1\left(\delta_{\mu\nu}\delta_{\rho\sigma} + \delta_{\mu\rho}\delta_{\nu\sigma} + \delta_{\mu\sigma}\delta_{\nu\rho}\right) + \lambda_2\delta_{\mu\nu\rho\sigma}, \quad (32)$$

$$\lambda_{ijkl}^{(2)} = \lambda_3\left(\delta_{ij}\delta_{kl} + \delta_{ik}\delta_{jl} + \delta_{il}\delta_{jk}\right) + \lambda_4\left(d_{ijm}d_{mkl} + d_{ikm}d_{mjl} + d_{ilm}d_{mkj}\right), \quad (33)$$

$$\lambda_{\mu\nu ij}^{(c)} = \lambda_5\delta_{\mu\nu}\delta_{ij}, \quad (34)$$

where $\lambda_{\mu i\nu j}, \lambda_{\mu ij\nu}\cdots$ are fixed by symmetry. From

$$\beta_{IJKL} = \beta_1\left(\delta_{\mu\nu}\delta_{\rho\sigma} + \delta_{\mu\rho}\delta_{\nu\sigma} + \delta_{\mu\sigma}\delta_{\nu\rho}\right) + \beta_2\delta_{\mu\nu\rho\sigma} + \beta_3\left(\delta_{ij}\delta_{kl} + \delta_{ik}\delta_{jl} + \delta_{il}\delta_{jk}\right) \quad (35)$$

$$+ \beta_4\left(d_{ijm}d_{klm} + d_{ikm}d_{jkm} + d_{ilm}d_{jkm}\right) + \beta_5\left(\delta_{\mu\nu}\delta_{ij} + \cdots\right), \quad (36)$$

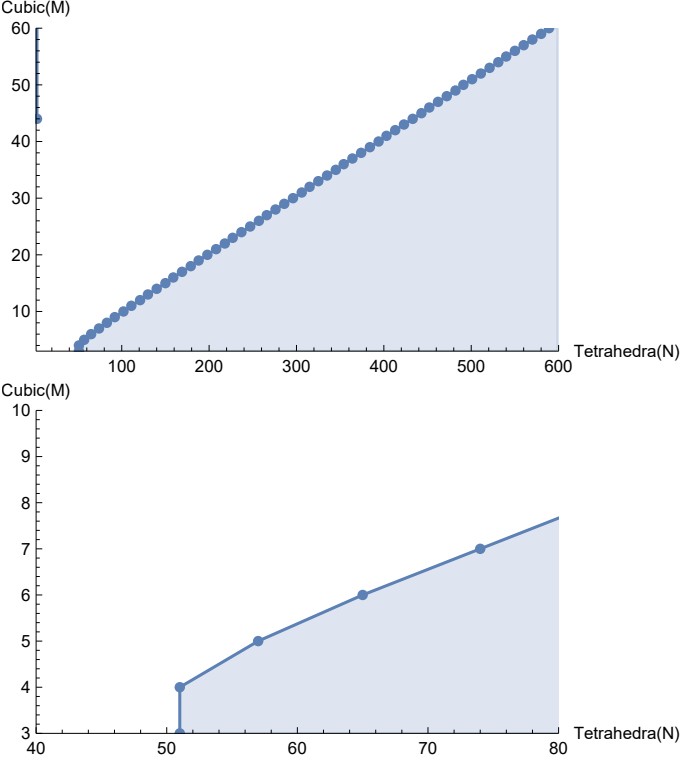

Figure 2: Cubic(M)×Tetrahedra(N). The right panel is a zoom in of the left panel plot. Notice that $N = 3$ and $M \geq 44$ is allowed, as indicated by the straight line close to the N-axis.

and (9), we get

$$\beta_1 = -\lambda_1 + 2\lambda_1\lambda_2 + \lambda_1^2(M+8) + \lambda_5^2 N\,, \tag{37}$$

$$\beta_2 = -\lambda_2 + \lambda_2\left(12\lambda_1 + 3\lambda_2\right)\,, \tag{38}$$

$$\beta_3 = -\lambda_3 + \lambda_5^2 M + \frac{4\lambda_4^2(N-1)(N+1)^4}{N^6} + \lambda_3\left(\frac{4\lambda_4(N-1)(N+1)^2}{N^3}\right) + \lambda_3^2(N+8), \tag{39}$$

$$\beta_4 = -\lambda_4 + \lambda_4\left(12\lambda_3 + \frac{3\lambda_4(3N-7)(N+1)^2}{N^3}\right)\,, \tag{40}$$

$$\beta_5 = -\lambda_5 + \lambda_5\left(2\lambda_1 + \lambda_2 + 2\lambda_3 + 4\lambda_5 + \lambda_1 M + \frac{2\lambda_4(N-1)(N+1)^2}{N^3} + \lambda_3 N\right)\,. \tag{41}$$

When $N$ is small, there are no negative mass eigenvalues no matter how large $M$ is. For every $M$, there is always an infinite family of Tetrahedral(N) CFTs such that they can couple together and the spontaneous symmetry breaking happens. The results are summarized in Fig. 2. The linear behavior of the plot at large N can be seen from a large N analysis, the details are summarized in Appendix A. Notice when $N = 3$, $M \geq 44$ is allowed. This agrees with the result in Fig. 1, after considering the group isomorphism Cubic(3)=Tetrahedral(3). These fixed points, as before, have at least one renormalization group flow unstable direction. When $M \neq 3$, it's always the Cubic group that gets spontaneously broken at finite temperature.

# 3 Discussion

It was noticed in [2] that the O(m)×O(N-m) fixed point has one RG flow unstable direction. Analyzing the stability matrix of several other CFTs proposed to exhibit all temperature SSB phases, we notice that they all share the same feature. These include the long range models in [8], and the 3+1 dimensional gauge theories proposed in [10]. Together with the two mass terms, this means that in real life or computer experiments, one needs to fine tune at least three parameters to reach the fixed point. This type of fixed points are accessible through numerical experiments [22].

Ideally though, one would like to find CFTs which can host an all temperature SSB phase, and whose operator spectrum contains as few relevant operators as possible. This could be achieved by looking at more general models. Notice we have only considered a sort of restricted set of fixed points, where two CFTs are coupled through their mass operators. One possibility is to consider scalar field theories with a single finite group $G$. One can choose $\vec{\phi}_1$ and $\vec{\phi}_2$ to be two irreducible representations of $G$, and then search for solutions to (6), such that $H_{ij}$ has no negative eigenvalues, while $M_{ij}$ has at least one. Preferably, the order of the finite group should be also small. This is a well defined group theoretical question. However, to work in this vast space of theories requires a more systematic study. The GAP system [23] for computational discrete algebra and in particular its small group library will be useful in searching for potential finite groups and their representations. For an application of GAP to study SUSY fixed points, see [24].

The beta functions we have used are one loop perturbative beta functions in $4-\epsilon$. It will be interesting to continue calculating higher loop corrections to shed light on the (non)existence of 2+1D CFTs whose SSB phase persists at all temperatures ($\epsilon = 1$). Our models are closely related to the models with non-trivial large N limit discussed in [25], as the Cubic(N) CFT at large N can be treated as N copies of the Ising model coupled together. The critical exponents of the Cubic(N) CFT can then be calculated by the diagrammatic method developed in [26] using the conformal data of the three dimensional Ising model from the numerical bootstrap [27–31]. The Cubic(M)×Cubic(N) CFTs which we discussed in Section 2.1 may be treated in a similar manner when $M$ and $N$ are both large (with $M/N$ fixed). Such an analysis could give evidence for the existence of 2+1D CFTs with an all temperature SSB phase, beyond the one-loop results of this paper. We leave this for future work.

One can also try to engineer a two dimensional lattice Hamiltonian by coupling $M + N$ copies of the quantum Ising model together. To start, let us consider the transverse field Ising model [32] on a square lattice,

$$H_{\text{Ising}} = -J \sum_{\langle ij \rangle} Z_i Z_j - h \sum_i X_i. \tag{42}$$

Here $Z_i$ and $X_i$ are the Pauli matrices. The symbol $\langle ij \rangle$ means that the sites $i$ and $j$ are nearest neighbors. It is well-known that as one tunes the parameter $h/J$, such a model goes through a second order phase transition, which is simply the 2+1 dimensional Ising CFT. Coupling M-copies of Ising models together, we get the quantum Cubic model

$$H_{\text{cubic}} = -J \sum_{a=1}^{M} \sum_{\langle ij \rangle} Z_i^{(a)} Z_j^{(a)} - J_2 \sum_{a \neq b} \sum_{\langle ij \rangle} Z_i^{(a)} Z_j^{(a)} Z_i^{(b)} Z_j^{(b)} - h \sum_{i,a} X_i^{(a)}. \tag{43}$$

Here the symbol $a$ enumerates the M copies of the Ising spin, that live on the same site. The size of the Hilbert space on each site is therefore $2^M$. One can also introduce other coupling terms, such as

$$-h_2 \sum_{i,a \neq b} X_i^{(a)} X_i^{(b)} \,,$$

as long as it preserves the Cubic symmetry. This model can be viewed as a 2+1 dimensional quantum version of the N-color [33] Ashin-Teller model [34]. See [35,36] for the study of the two-color quantum Ashin-Teller model in 1+1 dimensions. In the two dimensional parameter space, $(J/h, J_2/h)$, there should exist a line of critical points corresponding to the Cubic(M) CFT. We can now try to couple two Cubic Hamiltonians together,

$$H = H_1 + H_2 + H_3 \,, \tag{44}$$

with

$$H_1 = -J \sum_{a=1}^{M} \sum_{\langle ij \rangle} Z_i^{(a)} Z_j^{(a)} - J_2 \sum_{a \neq b} \sum_{\langle ij \rangle} Z_i^{(a)} Z_j^{(a)} Z_i^{(b)} Z_j^{(b)} - h \sum_{i,a} X_i^{(a)} \,,$$

$$H_2 = -J \sum_{\rho=1}^{N} \sum_{\langle ij \rangle} Z_i'^{(\rho)} Z_j'^{(\rho)} - J_2' \sum_{\rho \neq \sigma} \sum_{\langle ij \rangle} Z_i'^{(\rho)} Z_j'^{(\rho)} Z_i'^{(\sigma)} Z_j'^{(\sigma)} - h' \sum_{i,\rho} X_i'^{(\rho)} \,,$$

$$H_3 = -K \sum_{a=1}^{M} \sum_{\rho=1}^{N} \sum_{\langle ij \rangle} Z_i'^{(\rho)} Z_j'^{(\rho)} Z_i^{(a)} Z_j^{(a)} \,.$$

Other terms such as

$$-H \sum_i \left( \sum_a X_i^{(a)} \right) \left( \sum_\rho X_i'^{(\rho)} \right) \tag{45}$$

are also allowed as long as they preserve the Cubic(M)×Cubic(N) symmetry. Based on the field theory calculation, we conjecture that in the five dimensional parameter space $(J_2/J, h/J, J_2'/J, h'/J, K/J)$, there exist a two dimensional critical surface, corresponding to the Cubic(M)×Cubic(N) fixed points discussed in Section 2.1. In other words, one needs to fine tune three parameters to reach this critical surface. Changing the other two parameters corresponds to turning on irrelevant operators of the CFT, which will not drive the RG flow away from the critical surface. Given that we choose the size of the group properly, once we turn on temperature at this critical surface, the SSB phase persists at all temperatures. Since the size of our cubic groups are large, testing this conjecture might be challenging.

The SSB phases at all temperature are interesting for black hole physics [37–41] through the AdS/CFT correspondence [42–44], if the corresponding CFT has a dual AdS theory with the Einstein Hilbert action as the gravitational term. Our models, like the models in [2], are vector models whose large N limit will not have standard AdS duals. The single trace spectrum contains higher spin operators, but not conserved [45]. If the dual theory indeed exist, it should be a Vasiliev's higher spin theory [46,47] with deformations that breaks the higher spin symmetry.

# Acknowledgments

We thank Noam Chai, Anatoly Dymarsky and Eliezer Rabinovici for insightful discussion about thermal order in conformal field theories and thank Zheng Yan for valuable discussion about tetracritical phenomena.

**Funding Information**  JR is supported by the Huawei Young Talents Program at IHES. HZ is grateful to his advisor Chris Pope's support. HZ is supported in part by DOE grant DE-FG02-13ER42020 and the Mitchell Institute. PL acknowledges support from the DFG through the Emmy Noether research group "The Conformal Bootstrap Program" project number 400570283, and through the German-Israeli Project Cooperation (DIP) grant "Holography and the Swampland".

# A  Large N analysis

## A.1  Cubic(M) × Cubic(N)

We can now perform a large N analysis of the fixed points. More precisely, we take the $M \to \infty$ limit the ratio $N/M$ fixed. From the numerical results notice the coupling constants have the following large M behavior $\lambda_2 \sim \lambda_4 \sim \frac{1}{3} + O\left(\frac{1}{N}\right), \lambda_1 \sim \lambda_3 \sim \lambda_5 \sim O\left(\frac{1}{N}\right)$. Inspired by the above observation, we take the following ansatz

$$N = xM, \quad \lambda_1 = \frac{a_1}{M} + \frac{a_{11}}{M^2} + \cdots, \quad \lambda_2 = \frac{1}{3} + \frac{a_2}{M} + \frac{a_{22}}{M^2} + \cdots, \quad \lambda_3 = \frac{a_3}{M} + \frac{a_{33}}{M^2} + \cdots, \quad \text{(A.1)}$$

$$\lambda_4 = \frac{1}{3} + \frac{a_4}{M} + \frac{a_{44}}{M^2} + \cdots, \quad \lambda_5 = \frac{a_5}{M} + \frac{a_{55}}{M^2} + \cdots, \quad \text{(A.2)}$$

and do $\frac{1}{M}$ expansion for beta function, leading order equations are

$$\beta_1 = \frac{2a_5 a_{55} x + 8a_1^2 + 2(a_2 + a_{11})a_1 - \frac{a_{11}}{3}}{M^2} + \frac{a_5^2 x + a_1^2 - \frac{a_1}{3}}{M} + \cdots, \quad \text{(A.3)}$$

$$\beta_2 = \frac{3a_2^2 + 12a_1 a_2 + 4a_{11} + a_{22}}{M^2} + \frac{4a_1 + a_2}{M} + \cdots, \quad \text{(A.4)}$$

$$\beta_3 = \frac{2a_3(a_{33} x + a_4) + 8a_3^2 + 2a_5 a_{55} - \frac{a_{33}}{3}}{M^2} + \frac{a_3^2 x - \frac{a_3}{3} + a_5^2}{M} + \cdots, \quad \text{(A.5)}$$

$$\beta_4 = \frac{3a_4^2 + 12a_3 a_4 + 4a_{33} + a_{44}}{M^2} + \frac{4a_3 + a_4}{M} + \cdots, \quad \text{(A.6)}$$

$$\beta_5 = \frac{a_5\left(a_{33} x + 2a_1 + a_2 + 2a_3 + a_4 + 4a_5 + a_{11}\right) + a_{55}\left(a_3 x + a_1 - \frac{1}{3}\right)}{M^2} + \quad \text{(A.7)}$$

$$+ \frac{a_5\left(a_3 x + a_1 - \frac{1}{3}\right)}{M} + \cdots. \quad \text{(A.8)}$$

Solving the leading order equation, we can find

$$a_1 = \frac{1}{3}(1 - 3a_3 x), \quad a_2 = \frac{4}{3}(3a_3 x - 1), \quad a_4 = -4a_3, \quad a_5 = -\frac{\sqrt{a_3 - 3a_3^2 x}}{\sqrt{3}}, \quad \text{(A.9)}$$

with $a_3$ left free, indicating the existence of a large-N perturbative conformal manifold. If we include the sub-leading corrections, we find

$$a_1 = \frac{1}{6}, \quad a_2 = -\frac{2}{3}, \quad a_3 = \frac{1}{6x}, \quad a_4 = -\frac{2}{3x}, \quad a_5 = -\frac{1}{6\sqrt{x}}, \tag{A.10}$$

$$a_{11} = \frac{-3a_{33}x^2 + x + 2\sqrt{x} + 1}{3x}, \quad a_{22} = \frac{4\left(3a_{33}x^2 - x - 2\sqrt{x} - 1\right)}{3x}, \tag{A.11}$$

$$a_{44} = -4a_{33}, \quad a_{55} = 0. \tag{A.12}$$

Plugging them back to the mass matrix, we find the two eigenvalues

$$m_1 = \frac{1}{6}\left(3 - \sqrt{x}\right) + \cdots, \quad m_2 = \frac{1}{6}\left(3 - \frac{1}{\sqrt{x}}\right) + \cdots. \tag{A.13}$$

Apparently, we need either $x > 9$ or $x < 1/9$ so that one of the eigenvalues will be negative.

## A.2 Cubic(M)×Tetrahedral(N)

For the Cubic(M)×Tetrahedral(N) fixed points, we find a similar large N behavio in this case $\lambda_2 \sim \frac{1}{9} + O\left(\frac{1}{N}\right), \lambda_4 \sim \frac{1}{3} + O\left(\frac{1}{N}\right), \lambda_1 \sim \lambda_3 \sim \lambda_5 \sim O\left(\frac{1}{N}\right)$. We take our ansatz to be

$$N = xM, \quad \lambda_1 = \frac{a_1}{M}, \quad \lambda_2 = \frac{1}{9} + \frac{a_2}{M}, \quad \lambda_3 = \frac{a_3}{M}, \quad \lambda_4 = \frac{1}{3} + \frac{a_4}{M}, \quad \lambda_5 = \frac{a_5}{M}. \tag{A.14}$$

The leading order beta function becomes

$$\beta_1 = \frac{1}{M}\left(a_1^2 x - \frac{5a_1}{9} + a_5^2 + \frac{4}{81x}\right), \tag{A.15}$$

$$\beta_2 = \frac{1}{M}\left(\frac{4a_1}{3} + a_2 - \frac{1}{27x}\right) + \cdots, \tag{A.16}$$

$$\beta_3 = \frac{1}{M}\left(a_5^2 x + a_3^2 - \frac{a_3}{3}\right) + \cdots, \tag{A.17}$$

$$\beta_4 = \frac{1}{M}\left(4a_3 + a_4\right) + \cdots, \tag{A.18}$$

$$\beta_5 = \frac{a_5}{M}\left(a_1 x + a_3 - \frac{4}{9}\right) + \cdots, \tag{A.19}$$

Solving them gives

$$a_2 = \frac{1 - 36a_1 x}{27x}, \quad a_3 = \frac{1}{9}\left(4 - 9a_1 x\right), \quad a_4 = \frac{4}{9}\left(9a_1 x - 4\right),$$

$$a_5 = -\frac{\sqrt{-81a_1^2 x^2 + 45a_1 x - 4}}{9\sqrt{x}}, \tag{A.20}$$

with $a_1$ left free, indicating the existence of a large-N perturbative conformal manifold. Plugging them back to mass matrix, we find

$$m_1 = -\frac{\sqrt{-81a_1^2 x^2 + 45a_1 x - 4}}{9\sqrt{x}} + a_1 x + \frac{2}{9},$$

$$m_2 = -\frac{1}{9}\sqrt{x}\sqrt{-81a_1^2 x^2 + 45a_1 x - 4} + \frac{1}{9}\left(4 - 9a_1 x\right) + \frac{1}{3}. \tag{A.21}$$

Again, do the same sub-leading analysis, we find $x, a_1$ must satisfy the constraint.

$$
\begin{aligned}
&-5832a_1^3 x^{7/2}\\
&-324a_1^2 x^2\left(x\sqrt{-81a_1^2 x^2+45a_1 x-4}-\sqrt{-81a_1^2 x^2+45a_1 x-4}-15\sqrt{x}\right)\\
&-40x\sqrt{-81a_1^2 x^2+45a_1 x-4}\\
&+18a_1 x\left(13x\sqrt{-81a_1^2 x^2+45a_1 x-4}-7\sqrt{-81a_1^2 x^2+45a_1 x-4}-66\sqrt{x}\right)\\
&+28\sqrt{-81a_1^2 x^2+45a_1 x-4}+80\sqrt{x}=0\,.
\end{aligned}
\tag{A.22}
$$

Combining with (A.21 ), we can numerically find the bound $x>9.800000\cdots$.

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
