# Peer review of "Spontaneous breaking of finite group symmetries at all temperatures"

_SciPost Physics, doi:SciPost Phys. 14, 168 (2023)_

## Round 3 · Referee Report · Anonymous · 2022-11-23

Report

This paper demonstrates the existence of CFTs in $(4-\epsilon)$ dimensions where certain finite group symmetries are spontaneously broken at any nonzero temperature. It builds upon the ideas developed in the last few years where the question of symmetry non-restoration in the high temperature limit of QFTs has been recast as that of symmetry breaking at nonzero temperatures in CFTs. Since a CFT lacks any intrinsic scale, if any of its symmetries is broken at a nonzero temperature, it will remain broken at all nonzero temperatures, thereby providing an example of symmetry non-restoration. The novelty of this work lies in finding new examples of CFTs with finite group symmetries (cubic and tetrahedral symmetries) which exhibit this phenomenon. This is particularly important as the Mermin-Wagner-Hohenberg-Coleman theorem rules out the possibility of spontaneous breaking of a continuous symmetry in a 3-dimensional local QFT at any nonzero temperature. However, it imposes no such constraint on finite group symmetries. Therefore, I expect the new $(4-\epsilon)$-dimensional models introduced in this work to provide fresh clues for exploring whether persistent symmetry breaking is possible in 3-dimensional local conformal theories. For this reason I would recommend the paper for publication after a few minor revisions.

Requested changes

I suggest the following minor revisions:

1- I think the authors should clearly mention which of the two symmetry groups is broken in each of the examples that they study and also clarify if there is any general pattern.

2- The penultimate line in the first paragraph of the introduction should be

"Another interesting feature is that these systems have “inverted” phase diagrams near the CFT fixed point, with the disordered phase at low temperature, while the ordered phase is at high temperature."

instead of

"Another interesting feature is that these systems have “inverted” phase diagrams near the CFT fixed point, with the ordered phase at low temperature, while the disordered phase is at high temperature."

3- Based on the form of the potential given in eq. 1.4 and the couplings given in the first line of eq. 2.4, there should be an additional factor of 3 multiplying $\lambda_1$ in the Lagrangian of the cubic(N) theory given at the beginning of section 2.1 in page 4.

4- The symmetry group for the fixed points that are given in table 1 should be Cubic(60)$\times$ Cubic(4) rather than Cubic(4) $\times$ Cubic(60).

5- In the last term of the first line of eq. 2.7 (and the same of eq. 2.8), the index $\sigma$ is written thrice. I think a couple of them should be replaced by a dummy index.

6- In the third term of the second line of eq. 2.7 (and the same of eq. 2.8 and eq. 2.10), there is an extra index $j$ in one of the delta functions which should be removed.

7- The above two typos are also there in eq. 2.11. Moreover, in its present form, eq. 2.11 is inconsistent with eq. 2.10 . To make these two equations consistent $\beta_1$ and $\beta_2$ in eq. 2.11 should be exchanged with $\beta_3$ and $\beta_4$ respectively. Furthermore, in eq. 2.11 the indices ($\mu,\nu,\rho,\sigma,\kappa$) should be exchanged with the indices ($i,j,k,l,m$).

8- In the second last line of section 2 (page 7), the figure that is referred to should be Fig. 1 instead of Fig. 2.4.

---

## Round 3 · Referee Report · Anonymous · 2022-11-29

Strengths

1- The work is well-motivated.
2- The analysis is largely self-contained.

Weaknesses

1- There is a focus on a few specific examples but a lack of a general treatment of biconical-type models.
2- The different fixed points and the ranges of $M$, $N$ for which they exist as unitary fixed points are not discussed at all. This is particularly relevant when the tetrahedral theory is considered.

Report

In this manuscript the authors perform an analysis of biconical-type fixed points in the $\varepsilon=4-d$ expansion in the context of persistent symmetry breaking. More specifically, they consider models of the type $\text{CFT}_1\times\text{CFT}_2$ coupled via their quadratic invariants, with $\text{CFT}_{1,2}$ taken to have either hypercubic or hypertetrahedral global symmetry, and examine the possibility that these biconical-type CFTs have spontaneously broken global symmetry for arbitrarily high temperature. The authors indeed find that these models display persistent spontaneous symmetry breaking for a variety of $M,N$, thus enlarging the space of fixed points with such behavior in the $\varepsilon$ expansion. Since the symmetry groups considered are discrete, the authors' results may continue to hold all the way down to $d=3$.

The motivation for this work is clear, and the paper is well written and contains new results. Despite the fact that in the end this is a limited example-by-example analysis, it is still a valuable addition to the literature around persistent symmetry breaking and should be published. A minor addition before publication is requested below.

Requested changes

1- In section 2.2 the authors should mention that for $M=3$ or $N=3$ the corresponding CFT's symmetry group is the same as that of the cubic theory. This relation is mentioned in section 5.3 but in relation to the results of section 5.1. In section 5.2 this makes their examples of persistent symmetry breaking of the type $\text{Cubic}(3)\times \text{Tetrahedral}(\geq 51)$. A corresponding comment should then be added in section 5.3, for $M=3$, $N\geq 51$ there. It should perhaps be pointed out in the discussion that all their examples of persistent symmetry breaking involve at least one cubic CFT in the product $\text{CFT}_1\times\text{CFT}_2$.

---

## Round 4 · Referee Report · Anonymous · 2023-1-10

Report

With the additions and corrections by the authors, I think the manuscript can now be published.

---

## Round 4 · Referee Report · Anonymous · 2023-2-4

Report

The authors have adequately addressed the issues mentioned in my previous report and I recommend the paper for publication.

I would like to just point out a minor typo. In the last paragraph of section 2.1 (page 5) where the authors refer to table 1, the case that is mentioned should be Cubic(60)$\times$Cubic(4) instead of Cubic(60)$\times$Cubic(3).

---

## Round 4 · Author Response

This is the modified version. The requested changes from the referees have been taken care of.

---

## Round 4 · List of Changes

REPLY TO REVIEWER 1
1, As requested by referee 1, we added to sections 2.2 and section 2.3 the discussion of the isotropy of the Tetrahedral(3) and Cubic(3) groups. We also added in section 2.2 the comment that the persistent symmetry-breaking models we found involve at least one cubic group.
REPLY TO REVIEWER 12
1, We mentioned explicitly which of the two groups is broken at finite temperature, as suggested by the referee.2
2, Near the CFT point, our model does have a disordered phase at a lower temperature, while the ordered phase is at a higher temperature. We have further clarified this at the end of the first paragraph.
3, The problem of the missing factor of 3 in the Lagrangian of the cubic(N) theory has been fixed (we have deleted the Lagrangian in section 2.1 since it is not really necessary to mention it there)
4, The symmetry group for the fixed points that are given in table 1 should be Cubic(60)× Cubic(4), we thank the referee for pointing this out.
5,6,7 The typos of indices in eqn (2.7), (2.8), (2.10) and (2.11) have been fixed.
8, We have fixed the mistake in referring to Fig. 1 in Section 2.3

Resubmission 2205.13964v5 on 22 March 2023

---

## Round 5 · Author Response

List of changes
The typo spotted by referee 2 has been fixed.

---

## Round 5 · List of Changes

The typo spotted by referee 2 has been fixed.

---

## Editorial Decision

published